# Cosmic Ionizing Radiation: A DNA Damaging Agent That May Underly Excess Cancer in Flight Crews

**DOI:** 10.3390/ijms25147670

**Published:** 2024-07-12

**Authors:** Sneh M. Toprani, Christopher Scheibler, Irina Mordukhovich, Eileen McNeely, Zachary D. Nagel

**Affiliations:** 1John B. Little Center for Radiation Sciences, Department of Environmental Health, Harvard T.H. Chan School of Public Health, Boston, MA 02115, USA; stoprani@hsph.harvard.edu; 2Department of Environmental Health, Harvard T.H. Chan School of Public Health, Boston, MA 02115, USA; cscheibler@gmail.com (C.S.); irina.bnjmn@gmail.com (I.M.); emcneely@iq.harvard.edu (E.M.); 3Sustainability and Health Initiative (SHINE), Department of Environmental Health, Harvard T.H. Chan School of Public Health, 665 Huntington Avenue, Boston, MA 02115, USA

**Keywords:** DNA damage, DNA repair, flight, cancer, cosmic ionizing radiation, flight attendants, pilots

## Abstract

In the United States, the Federal Aviation Administration has officially classified flight crews (FC) consisting of commercial pilots, cabin crew, or flight attendants as “radiation workers” since 1994 due to the potential for cosmic ionizing radiation (CIR) exposure at cruising altitudes originating from solar activity and galactic sources. Several epidemiological studies have documented elevated incidence and mortality for several cancers in FC, but it has not yet been possible to establish whether this is attributable to CIR. CIR and its constituents are known to cause a myriad of DNA lesions, which can lead to carcinogenesis unless DNA repair mechanisms remove them. But critical knowledge gaps exist with regard to the dosimetry of CIR, the role of other genotoxic exposures among FC, and whether possible biological mechanisms underlying higher cancer rates observed in FC exist. This review summarizes our understanding of the role of DNA damage and repair responses relevant to exposure to CIR in FC. We aimed to stimulate new research directions and provide information that will be useful for guiding regulatory, public health, and medical decision-making to protect and mitigate the risks for those who travel by air.

## 1. Introduction

Air travel increases yearly in the United States (US), exposing approximately 43,000 pilots (out of approximately 351,000 globally) and 96,900 flight attendants (FA) (out of a global workforce of 400,000) to doses of cosmic ionizing radiation (CIR) much higher than those experienced at sea level [1,2,3]. Still, we have yet to fully understand CIR-related health risks. The National Council on Radiation Protection and Measurements (NCRP) has formally classified flight crews (FC), which include pilots and FA, as radiation workers [4], receiving an average annual effective dose of 3.07 mSv. This dose is equivalent to approximately 30 chest X-rays [5] and is five times the average dose (0.59 mSv) of US Department of Energy radiation workers [4]. Though significant, this exposure falls well within the United Nations’ Scientific Committee on the Effects of Atomic Radiation (UNSCEAR) definition of low-dose ionizing radiation (IR) (<100 mSv) [6]. The levels and composition CIR vary depending on both the altitude and latitude [7]. Circumpolar flights operating at cruising altitudes of 35,000 feet or above increase the exposure to CIR of FC. The earth’s magnetic field and atmosphere attenuate CIR, but to a lesser degree at higher altitudes and more polar latitudes [8]. Solar particle events (SPEs) occur frequently and can modulate the magnitude of the exposure to CIR of FC [9]. The complex interplay among the many factors influencing CIR underscores a need for detailed dosimetry and further investigation of the biological effects of this exposure.

While several large studies have documented the health risks associated with working as FC [10,11,12,13], they remain an understudied occupational cohort with respect to understanding the mechanisms underlying the biological effects of exposure to CIR [14,15]. Mounting evidence from several epidemiological studies has indicated elevated rates of melanoma skin cancer, non-melanoma skin cancer, brain/central nervous system cancer, and breast cancer have been associated with cumulative exposure to CIR in FC [11,12,16,17,18], but some studies have not found a relationship between cumulative exposure and cancer risk [11,19,20]. Efforts to link excess cancer risk in FC exposed to CIR have been hampered by a lack of accurate in-flight CIR measurements and the heterogenous composition of CIR, which represents high linear energy transfer (LET) radiation and precludes a direct comparison with the exposure to low-LET radiation experienced by other radiation workers. Understanding the effect of exposure to CIR alone in FC is challenging, as CIR is difficult to measure precisely, and the exposure occurs in concert with exposure to a complex mixture of stressors and chemicals from the in-flight environment [11,12,15]. In addition, FC generally have higher health standards for employment (particularly pilots), because of regular “fitness for duty” or “fit to fly” assessments directed at weeding out individuals with serious health conditions that may jeopardize safety and performance. This constraint, or study bias, is known as the “healthy worker effect” [21]. The existing technological constraint of measuring real-time exposure to CIR for an individual member of the FC and the difficulty finding a near to similar control individuals for a comparison with landscape of FC exposure during air travel can contribute to problems in corroborating data with a biological endpoint. 

Compromised genomic integrity is a common thread that links many types of exposure encountered by FC, which may explain their excess cancer risk. Among these exposures, CIR is particularly interesting because exposures to this DNA-damaging agent in FC are estimated to be as much as 2.05-fold higher than in the general population (1.0 mSv) [4,22]. Various DNA lesions are expected to result during air travel, albeit at very low levels, given the extremely low dose rate of CIR. Furthermore, the extent of genomic damage during air travel increases with altitude, alongside with increases in both dose and composition of CIR [4,23,24]. By contrast, endogenous cellular processes contribute to a baseline level of oxidative DNA damage that includes approximately 70,000 DNA lesions per cell per day [25]. It is important to consider that while endogenously induced DNA lesions far outnumber CIR-induced DNA lesions [26], CIR can induce complex DNA lesions that are more challenging to repair and are more likely to cause mutations or cell death [27]. Thus, there is a need to understand CIR-induced DNA damage, its mechanisms of repair, and its biological consequences. 

Multiple molecular pathways are involved in the repair of radiation-induced DNA damage. Variation in the efficiency of DNA repair pathways can govern an individual’s sensitivity to a mutagenic exposure, explaining their susceptibility to cancer. Accumulating unrepaired DNA lesions can contribute to mutations and cellular senescence, resulting in various adverse health outcomes [28,29,30]. In this review, we present the current understanding of how DNA damage and repair in the context of exposure to CIR may explain, at least in part, the elevated risk of some types of cancer in FC. This review also outlines current knowledge regarding exposure to CIR during flights, discusses the available methods for detecting CIR, and provides a brief overview of the regulatory guidelines. We also present our perspectives on gaps in our understanding of CIR’s effects and what will be needed to understand the biological mechanisms in the field of DNA damage and repair underlying the excess cancer in FC to help develop appropriate radioprotection protocols for air travel [31]. 

For this review, the methodology is outlined as follows. In Section 2, we familiarize the reader with the concept of CIR, detailing its detection methods and current regulatory measures. To collect up-to-date information up to May 2024, we performed searches on Google and the PubMed website. Keywords such as “air travel”, “cosmic radiation”, “flight/airline/commercial pilots”, “regulation”, and “CIR dosimetry” were used to ensure comprehensive coverage. All relevant and updated data were included. To compile regulatory doses of radiation for occupational workers, we referenced to published and publicly available policy letters or guidelines from authoritative websites. In Section 3 and Section 4, we dive into finding the disperse and scare research work carried out by researchers in field of DNA damage and repair on the FC cohort. We searched for the words “flight attendants”, “flight/airline/commercial pilots”, “DNA damage”, “DNA repair”, “CIR”, “cosmic radiation”, and “cosmic ionizing radiation” on PubMed up to May 2024. Additionally, we focused on studies with a sample size greater than 15 that supported their conclusions with statistical analyses. 

## 2. Current Understanding of Exposure to CIR during Flight Travel

### 2.1. Exposure to CIR during Flight

CIR comprises solar cosmic radiation (SCR) created by solar activity and galactic cosmic radiation (GCR) that originates from outer space and is caused by distant events such as supernovas. SCR is characterized by continuously emitted radiation, known colloquially as the solar wind, and sporadic solar particle events (SPEs) [7]. The components of CIR also shift with the intensity of SCR, oscillating on an 11-year solar cycle. SPEs occur frequently but randomly with variable magnitudes, leading to short-term fluctuations in exposure to CIR [9]. For instance, as per the US National Oceanic and Atmospheric Administration (NOAA), a recent blackout in Southeast Asia and Australia on 17 April 2022 and another in Australia and New Zealand on 7 November 2022 were caused by solar flares, leading to significant shortwave radio blackouts with a radiofrequency below 30 mHz. No measurement of exposure to CIR (GCR + SCR) doses was made, underscoring the need for accurate dosimetry and experimental data regarding exposure to CIR in combination with SPEs. The impact of SPE is yet to be fully understood or accounted for to estimate the exposure of FC, which was beyond the scope of this study. Neutrons are the predominant component of CIR (~55%) at flight altitudes, with the remainder comprising both low and high linear energy transfer (LET) radiation [32] (Figure 1). The currently available aircraft types and materials cannot easily block the large portion of neutron radiation entering the cabin environment [33,34,35,36]. Circumpolar flights at cruising altitudes of 35,000 feet or above are significantly impacted by exposure to CIR due to lower protection being imparted by the Earth’s atmosphere and magnetic field [8]. With each 4500-foot increase in altitude, there is an estimated doubling of exposure to CIR. At the polar latitudes, the levels of radiation are approximately twice as high as at the equator [7]. The heliosphere acts as a giant shield, protecting the planet from GCR, but this protection varies inversely with solar activity [37,38,39]. The intensity of the exposure to GCR increases with altitude up to a peak level of approximately 65,000 feet, known as the Regener–Pfotzer maximum, where cosmic rays impact the Earth’s atmosphere and initiate the cascade of secondary particles and photons that contribute to increased levels of CIR [40]. However, recent studies using new dosimetry techniques have found evidence of increasing levels of CIR beyond this maximum [41] (Figure 2). At flight altitudes (35,000 ft and above), in addition to CIR, FC are potentially subject to terrestrial gamma ray flashes associated with thunderstorms and lightning [14,42]. FC have been classified as “radiation workers” by various regulatory agencies, such as the International Commission on Radiological Protection (ICRP) in 1990, the Federal Aviation Administration (FAA) in 1994, and the NCRP in 2009, and some regulatory agencies have imposed limits on exposure to CIR Table 1. In contrast to workers exposed to a single radiation source, FC are exposed to CIR and SPEs, and may also be exposed to non-CIR radiation sources such as radioactive cargo, airport security scanners, medical imaging related to occupational medical surveillance requirements, and ultraviolet radiation, but the magnitude and biological impact of these exposures are still unexplored in FC. Regulatory guidelines have frequently been updated as our understanding of the health effects of radiation has advanced (Figure 3). Taken together, the complex factors influencing levels of CIR underscore the need for accurate and detailed dosimetry for assessing the health effects of exposure to CIR during flight and informing regulatory guidelines. 

### 2.2. Methods for Measuring or Estimating Exposure to CIR during Air Travel

Although FC have been exposed to CIR since the advent of commercial flight, several challenges have confronted efforts to assess exposure to CIR and understand the impact of CIR on FC health. Data regarding exposure to CIR to FC are limited because measurements of CIR are not routinely carried out on commercial airliners. Historically, measurements of CIR taken during commercial flights have been primarily carried out onboard the Concorde aircraft, as it was the only commercial aircraft equipped with radiation dosimeters. This was due to its unique supersonic cruising altitude (60,000 feet), which motivated the use of onboard dosimetry to monitor inflight CIR for radioprotective purposes. Early data collected aboard the Concorde yielded CIR dose rates of 12–15 μSv per hour for supersonic flight. Today’s standard commercial aircraft yield measurements of 4–6 μSv per hour on long-haul subsonic flights and 1–3 μSv per hour on short-haul subsonic flights [46]. For comparison, the average US resident receives an estimated annual exposure to background radiation of 3.1 mSv from sources of ground-level cosmic radiation, surface soil radionuclides, radon, and radionuclides in the body [5,22]. The average annual dose of exposure to CIR in FC ranges from 3.07 to 6.0 mSv, based on flying hours and cruising altitude [7,47], with some cases of FC having exposures up to 9.0 mSv [48]. Individual lifetime cumulative doses have been estimated at >100 mSv [49]. Notably, cumulative exposure to CIR levels calculated using Concorde-based approaches often exceed FAA- or ICRP-informed guidelines created to protect workers and the public from possible radiation-induced health effects in other contexts [45,46,49]. 

Limited direct measurements of exposure to CIR create a major barrier to studying its biological effects in FC. The ideal approach would be to take individual measurements during every flight that they take, but this is not currently feasible. However, directly measuring levels of CIR at altitude can be difficult. Measurement devices have historically been cumbersome due to the size of dosimeter and limited in their ability to detect the full spectrum of radiation that comprises CIR. Other than in the case of the Concorde, aircraft manufacturers have not prioritized the placement of static radiation dosimeters into aircraft designs. Without onboard equipment, some of the experimental measurements have been performed with temporary dosimeters or individual aircraft details that are unable to measure the cumulative dose received by FC over longer time periods [7,50,51,52]. 

Although indirect measurements using computational models that estimate exposure to CIR based on the flights’ characteristics (origin, destination, route, date, etc.) have been used, these platforms vary in their estimations, depending on which component of CIR is considered [43]. Because exposure to CIR depends on the route, duration, latitude, and altitude of the flight, estimates of exposure to CIR need to be based on the actual flight mix encountered by FC, but these tend to be highly variable. Exposure to CIR can also be highly variable due to the solar cycle and SPEs. Together, these factors create a need for additional research aimed at obtaining accurate, individualized dosimetry and the analysis of potential biomarkers of exposure to CIR to determine CIR-related health risks in FC. 

## 3. Interindividual Variation in Response to Exposure to Radiation 

Some have postulated that low doses of exposure to radiation may have beneficial effects on human health [53,54,55]. Since FC are constantly exposed to chronic low doses of radiation, according to this theory, it is possible that other genotoxic exposures cause the excess cancers seen in this population, and the FC are, in fact, less susceptible to health risks from exposure to radiation due to an adaptive response (AR), according to epidemiological findings associating the cancer risk in relation to CIR [11,56]. It can further be speculated that there might be heterogeneity in individual responses to CIR due to differences in the AR among FC. Such differences have been reported in the response of human lymphocytes to radiation [57], which could further complicate the biological outcomes of exposures to CIR in FC [58]. Although some studies have found an AR in human subjects, the results have been inconclusive, debatable, and ambiguous [53,57,59]. Among those that saw an AR, the magnitude of the response varied from person to person [59], further illustrating the complexity of interindividual differences and highlighting the challenges of detecting possible ARs in FC. The only study investigating ARs in FC reported that lymphocytes had higher rates of spontaneous chromosomal aberrations and greater chromosomal sensitivity to bleomycin [60], opposite to what would be expected in the case of an AR. Given the inconsistent findings on the AR across human studies, further research is needed to establish the existence of this protective mechanism. Furthermore, assessing risks that are relevant to the chronic low-dose exposure of mixed (high- and low-LET) radiation in FC are challenging, as most of the biological effects have been evaluated on exposure to individual components of CIR at much higher doses than those experienced during flight.

During development, many cells are actively dividing and thus more susceptible to DNA damage, raising concerns about the potential health effects of prenatal exposure to radiation. The gestational age of the fetus at the time of exposure to radiation significantly influences the risk for adverse health effects in later stages of life [61]. Specifically, DNA damage and chromosomal aberrations, changes in ancillary cellular processes, growth deformities, and cognitive defects have been observed upon prenatal exposure to radiation [62,63,64,65]. The study by Grajweski et al., incorporated the subjects’ occupational and recreational flight history for the 6 months before pregnancy and during pregnancy, finding mean effective doses of 1.8 mSv before pregnancy and 0.36 mSv during the first trimester in the FA cohort. The study also observed a near-significant positive estimate in spontaneous abortion (OR 1.7, 95% CI: 0.95, 3.2) in FA with higher estimated exposure to CIR during the first trimester, suggesting a 70% increased risk above 0.1 mGy [66]. In a further assessment of the complex factors confounding the flight environment, the study by Grajewski et al. also evaluated other exposures experienced by FAs and found an association of first-trimester miscarriage with disruption of the circadian rhythm (OR 1.5, 95% CI: 1.1, 2.2) and high physical job demands (OR 2.5, 95% CI: 1.5, 4.2) [66]. Overall, across reproductive epidemiological studies, both the direction of the associations and the significance of the findings are mixed. However, at the least, there is enough evidence to approach exposure to CIR and pregnancy conservatively, and there appears to be trending evidence to argue that there is increased risk of adverse reproductive effects in FA associated with their employment as FC (and CIR as a proxy), and potentially in pregnant air travelers. Further studies with well-designed estimations of CIR are needed to confirm the suspected health risks. The limitations of the conclusions drawn from these studies include the small number of studies using a variety of outcomes and studies not accounting for estimations of CIR in their designs other than by occupation. Another limitation is that case definitions vary by study, with different thresholds for the inclusion of cases for outcomes such as intrauterine fetal demise and spontaneous abortion. Several studies that utilized questionnaire responses were limited by a low study response rate [67,68], while other studies were weakened by low numbers of cases in exposed groups [69]. Furthermore, in these analyses, the study designs and estimates of the effects were mixed, including the use of small cohorts, having different comparison groups, and variable statistical reporting methods. The majority of the literature has evaluated Western populations, and the one occupational cohort exception that evaluated the risk in Chinese FA may be confounded by a Chinese regulation that requires FA to be removed from work upon confirmation of pregnancy [70].

Given the importance of DNA repair in mitigating the biological effects of radiation, interindividual differences in DNA repair capacity (DRC) may be expected to play a role in susceptibility to the potential health effects of exposure to CIR. Several factors govern variations in DRC. These include genetics, sex, age, lifestyle, environmental exposures, etc. [71,72]. Interindividual variation in DRC for radiation-induced damage has been reported among apparently healthy individuals [57,73,74]. Genome-wide association studies (GWAS) have also linked DNA repair gene variants with various cancer risks in healthy individuals [75]. There is a need for an improved understanding of CIR-induced DNA damage and the interindividual differences in DRC, which could help to explain the elevated cancer risk observed in FC. Identifying biomarkers for environmental exposure and disease is pivotal for population science. To accounting for low levels of damage detected upon exposure to CIR in contrast to endogenous DNA lesions, there is a need for more sensitive and specific assays to predict the near to real-time biological effects induced by exposure to CIR in FC.

## 4. DNA Damage and Repair Mechanisms Associated with Exposure to CIR in FC

Although direct, accurate CIR dosimetry is extremely limited in FC, indirect estimates of exposure based on modeling have afforded a strategy for linking CIR with the health outcomes and biomarkers of exposure to radiation [52,76,77,78,79,80,81,82]. In this section, we discuss the findings of these studies, with an emphasis on DNA damage and repair mechanisms.

### 4.1. Markers of DNA Damage and Genomic Instability in FC That Are Consistent with the Expected Biological Effects of Exposure to CIR 

This section of the review summarizes the available data supporting a model wherein the elevated cancer risk observed in FC can be explained, at least in part, by an underlying molecular mechanism involving CIR-induced DNA damage. Since direct experimentation with actual CIR is not feasible, evidence for potentially harmful occupational exposure to CIR among FC has come from investigations of biomarkers in response to exposure to sources of terrestrial radiation meant to model the qualities of CIR. Although these types of radiation cannot exactly recapitulate the properties of bona fide CIR, they have provided an important model for how CIR may be expected to affect biological systems.

Each day, cells experience >100,000 spontaneous DNA lesions, which include base damage, oxidative damage products such as 8-hydroxy-2-deoxyguanosine (8oxoG), single-strand breaks (SSBs), and double-strand breaks (DSBs) [83]. These endogenously produced DNA lesions are usually resolved efficiently by the cellular DNA repair machinery. However, defects in DNA repair can lead to genomic instability [84], cellular dysfunction, and cell death, underlying age-related diseases such as cancer and neurodegenerative disorders [85]. The same types of DNA lesions can be induced by IR, including CIR, but they are more likely to occur as clusters of lesions comprising more than one type of damage [86,87,88]. When two or more isolated lesions occur within 10–20 bp (i.e., ~1–2 helical turns of the DNA), the damage is referred to as a clustered DNA lesion [83,89]. Clustered lesions may include complex DSBs and small DNA fragments arising from multiple closely spaced DSBs [24,90].

An extremely low dose rate relative to background exposure makes it difficult to detect CIR-induced damage. However, this type of radiation has properties that distinguish it from other types of radiation and may enable specialized strategies for detecting its effects on the genome. CIR includes high-LET radiation, which differs from low-LET radiation, such as X-rays, in terms of the density of ionization along the track of radiation [91] and the relative biological effectiveness (a relative measure of DNA damage caused by a radiation per unit track of energy deposition in a biological tissue) [92], which govern the types of DNA damage induced [93]. Complex clusters of oxidative DNA lesions are produced to a greater extent by high-LET IR than by low-LET IR and, as a result, have more harmful consequences [87,94]. For instance, human cells exposed to 1.0 Gy ^56^Fe ions produce 20–50 DNA fragments of <1000 base pairs, about 30 times higher than 1.0 Gy of γ rays [90]. DNA damage induced by heavy ions is more efficient in inducing cellular senescence [95], premature chromosomal condensation (biodosimetry to detect interphase chromosomal damage) [96], the formation of micronuclei (a biomarker of IR-induced unrepaired DSBs during the cell cycle) [97], and γ-H2AX (a biomarker of DSBs) [98] compared with high doses of low-LET IR. Interestingly, and consistent with the induction of a different spectrum of DNA lesions, the mutational signatures induced by high-LET IR are distinct from those induced by low-LET IR [99]. The studies discussed in this section involve much higher levels of DNA damage than would be expected upon exposure to CIR in FC. According to the linear no threshold (LNT) model, it is assumed that the risk associated with higher doses can be extrapolated to lower doses. However, alternative dose–response relationships have been proposed, including some that depend upon the details of the of exposure, including the dose, dose rate, and type of radiation [100].

The biological effects of CIR’s component particles on cells depend significantly on the dose rate. One study quantitated the level of γ-H2AX foci following exposure to a low dose rate (0.015 Gy/min) vs. a high dose rate (0.400 Gy/min) of neutrons at several total doses (0.125 Gy to 2.0 Gy) in human peripheral blood mononuclear cells (PBMCs). In this study, averaged over all doses, 40% greater induction of γ-H2AX foci was observed after exposure to a high neutron dose rate compared with exposure to a low dose rate. Levels of the foci decreased 24 h after irradiation, and the foci remained significantly higher than the background levels irrespective of the neutron dose rate [101]. This indicated that the dose rate is a factor that may need to be considered for estimations of cancer risk from exposure to neutrons, which are the most abundant component of CIR.

A very limited number of studies have directly measured the biomarkers of DNA damage in FC. A study on 44 male airline pilots compared with 36 factory workers was conducted to determine whether indirect estimates of CIR can correlate with the levels of 8oxoG in blood and urine samples [37]. The study showed that the levels of 8oxoG were higher in pilots, though it was not possible to establish a CIR dose–response relationship. It has also been hypothesized that interactions with non-ionizing radiofrequencies (e.g., in-flight Wi-Fi) and low-frequency electromagnetic fields aboard aircraft could augment the deleterious effects of CIR by increasing cellular oxidative stress [102,103,104]; however, adverse health effects from non-IR exposure are inconsistent across the literature [105]. A study using comet assays to compare civil FC flying long-haul routes versus matched ground staff showed a non-significant trend toward higher levels of basal DNA damage among FC (measured as oxidative damage, SSBs, and DSBs) [106]. A recent study conducted with smaller sample sizes, considering seasonal variations, revealed a significant increase in DNA damage in airline pilots compared with 40 male office workers matched on the basis of age and tenure of service (requiring at least 5 years) [107]. When comparing exposure to high-LET radiation versus exposure to low-LET radiation, divergent patterns of DNA damage, gene expression, mobilization of repair proteins, cytokine activation, and remodeling of the cellular microenvironment were observed [108]. Furthermore, some particle components of CIR, which have not been studied as thoroughly as photon radiation, also have distinctive properties concerning their biological effects [109], as well as the types of DNA damage they produce, as detailed above. Sensitive assays capable of detecting these rare, distinctive types of DNA damage are needed to investigate the biological effects of CIR.

Several studies using cytogenetic analyses (chromosomal aberrations, formation of micronuclei, sister chromatid exchanges) have yielded mixed results with regard to evidence of CIR-induced damage in FC. Chromosomal aberrations, formed by inaccurately repaired or unrepaired DSBs [110] in human lymphocytes, have been considered a reliable indicator of low-dose exposure to radiation [6]. They can be observed several years after exposure to radiation in PBMCs, such as in atomic bomb survivors [111,112]. Some FC studies have found significantly elevated levels of chromosomal aberrations versus the selected control group [76,77,78,107]; however, others found no difference [79,80]. The different findings of these studies may be due to differences in the studies’ designs, including the selection of the control group, and whether a sufficient number of cells were scored to detect an expected small increase in these relatively rare cytological events. Significantly higher levels of chromosomal aberrations and micronuclei were observed when comparing the Concorde pilot group (an estimated mean dose per year from 11 to 37 mSv, depending on radiation weighting factor of neutrons) versus controls who were matched for age, health, and socioeconomic status [76]. Similar statistically significant differences were also observed for civilian pilots and the cabin crew of subsonic aircraft (*n* = 192; 120 males and 72 females) compared with non-flying healthy volunteers (*n* = 55; 24 males and 31 females) [77]. On the other hand, a different study showed no difference between 83 airline pilots versus 50 comparison subjects (mean age: 47 versus 46 years, respectively) [80]. This underscores the point that the choice of control group is a critical factor. In addition, the parameters used for scoring chromosomal aberrations can play an important role in the outcome. For example, interlaboratory variations in recognizing, rejecting, and classifying various types of chromosomal aberrations and setting different thresholds for scoring have been documented [113]. The frequency of chromosomal aberrations is expected to be directly proportional to exposure to radiation. Yet, while some FC studies have yielded findings that are consistent with this expectation [79,80], others have not [78]. Significantly higher levels of chromosomal translocations per cell were observed in long-term male pilots (healthy, non-smoking, aged 40–60 years, recruited by a single airline) versus non-pilot controls (aged-matched, no history of frequent flying) [78]. However, the number of translocations per cell showed no dose–response relationship among the pilots [78]. Similarly, the study by Grajewski et al. in 2018 [52] observed no association in the frequency of translocation and the estimated absorbed dose in all types of flying male pilots. By contrast, linear relationships between cytokinesis-blocked micronuclei and the average annual effective CIR dose of radiation received or the average annual flying hours of FC were observed in PBMCs [81]. The DNA damage underlying cytogenetic abnormalities in FC could potentially be attributed to other in-flight exposures. Nevertheless, studies that have found elevated levels of chromosomal aberrations, particularly those finding a dose–response relationship with estimated exposure to CIR, are consistent with genomic instability induced by CIR.

Animal- and cell-culture-based models have provided insights into the mechanisms that may be involved in the biological effects of CIR. These models remove the complexity inherent in population settings and enable more detailed investigations of the impact of CIR or its components in a cell- or tissue-specific manner that cannot be achieved using the biological samples usually available from human subjects. Exposure to CIR is expected to have tissue-specific effects on the human body. It is well established that the biological effects of IR are tissue-dependent (Figure 4) [114,115], partly due to tissue-specific sensitivity to the induction of apoptosis [116]. Thus, there is a need for additional studies to understand the underlying mechanisms. Li et al. reported that exposure to ^48^Ti ions (one of the components of GCR) induced deletions in lung-derived epithelial cells in vitro when DSBs were induced in regions of the genome that have flanking short microhomologies that promote error-prone alternative end-joining pathways. In addition, in bronchial epithelial cells, they found an increase in chromosomal rearrangements, which are associated with an increased risk of lung cancer [88]. Persistent epigenetic effects, such as alterations in DNA methylation, have also been reported in lung cells exposed to components of GCR [117].

Despite important advances, our understanding of the impact of exposure to CIR on human health remains limited and warrants further investigation. DNA repair mechanisms remove the damage induced by CIR. Dysregulation of the DNA repair pathways promotes genomic instability and may underly the elevated cancer risk observed in FC. The next section delineates the major DNA repair pathways, with an emphasis on their roles in repairing the DNA damage expected to be induced by CIR. The challenge of measuring CIR-induced damage has hampered efforts to study its health effects, but, given the central importance of DNA damage and repair for cancer [122], CIR-induced genomic instability provides a biologically plausible explanation for the elevated cancer risk in FC that needs to be further investigated.

### 4.2. DNA Repair Mechanisms Involved in the Cellular Response to the Damage Induced by Components of CIR 

Genomic instability is a hallmark of cancer and drives radiation-induced carcinogenesis [123,124]. To maintain genomic stability, cells use numerous DNA repair mechanisms [125]. Although it has not been feasible to study these mechanisms directly in the context of CIR, various laboratory-based model systems have been used [126,127,128]. Each component of CIR is expected to induce a distinct spectrum of DNA lesions, each recognized and processed by a specific DNA repair mechanism. To understand the implications of this complexity, many groups have investigated the DNA damage response (DDR) in human PBMCs exposed to various doses of alpha particles, X-rays, and mixed beams of radiation [129]. For example, the expression of DDR genes such as ataxia telangiectasia mutated protein (*ATM*), tumor protein 53 (*P53*), and DNA protein kinase catalytic subunit (*DNA PKcs*) have been reported to be induced to a greater degree when cells are exposed to comparable doses of mixed beams versus pure high-LET alpha particles or pure low-LET X-rays [129]. This suggests that mixed high- and low-LET radiation may present a greater challenge to genomic integrity. Experiments using neutron sources in vitro circumvent some of these challenges and have provided insights into the potential genotoxic effects of CIR. For instance, compared with mock-irradiated controls, a significant increase in the frequency of micronuclei, levels of DNA damage, and γ-H2AX foci was observed after irradiation with neutrons from a ^241^Am-^9^Be source at doses as low as 9.0 mGy [130]. In addition, consistent with the induction of a DDR, increases in the levels of *GADD45A*, *CDKN1A,* and *PARP1* transcripts were reported at 4 h post-irradiation with low doses of neutrons in human resting PBMCs [130]. The molecular mechanisms of DNA repair have been extensively studied for higher doses of IR. However, there is a gap in our understanding of the biological effects and health risks associated with exposure to low-dose or low-dose-rate IR and complex mixtures of exposure to low- and high-LET radiation, as seen in CIR.

In the remainder of this section, we summarize the available literature on the possible role of CIR-induced DNA damage and DNA repair pathways in the health outcomes of FC. Since very little work has been carried out with actual CIR-induced DNA damage, most of this work relies on model systems and assumptions based on our understanding of the biological effects of IR. IR produces a variety of types of DNA damage. For example, 1.0 Gy of gamma rays induces thousands of base lesions and SSBs and about 40 DSBs per human cell [26]. Relative to low-LET photon radiation, the high-LET components of CIR may induce a higher proportion of DSBs [131], which are mainly repaired by the **non-homologous end joining (NHEJ)** and **homologous recombination (HR)** pathways. Defects in the activity of NHEJ and HR have been associated with chromosomal aberrations, immune dysfunction, clinical radiosensitivity, and elevated cancer risk [132,133]. The presence of one or more DNA lesions at or near the ends of DSBs requires additional factors for repair by the NHEJ pathway [134], and the need for DNA end-processing increases the possibility of slower and alternative error-prone DNA repair mechanisms [135,136]. A linear dose–response relationship indicates that a single ionization track gives rise to most DSBs for both high- and low-LET radiation [26], but the dense ionization tracks associated with high-LET radiation are more likely to give rise to complex lesions that are repaired slowly compared with those induced by photons [137]. In addition to directly induced DSBs, radiation can lead to DSBs by indirect means that remain incompletely understood. Genomic instability can also occur in the progeny of irradiated cells and is termed delayed hyperrecombination [138,139]. Several-fold increases in delayed hyperrecombination that last for multiple weeks have been reported in response to low-LET X-rays or high-LET carbon ion radiation in mice [138]. In further support of the notion that for a more prominent role in DSB repair of the DNA damage induced by high-LET radiation, some studies have demonstrated HR-deficient cell lines that have greater sensitivity to proton radiation than low-LET photons with the same energy [140]. **Microhomology-mediated end joining (MMEJ)** (a mutagenic DSB repair mechanism that uses microhomologies flanking the site of the break to guide repair) [141] also contributes to the repair of radiation-induced DSB; the radio-sensitizing effects of MMEJ deficiency depend upon the type of cell and the genetic context [142,143]. However, the role of NHEJ, HR, and MMEJ still remains unexplored in repair of CIR-induced damage in the FC cohort.

SSBs, abasic sites, and oxidative DNA damage, the most abundant types of DNA lesions produced by IR, are primarily repaired by the **base excision repair (BER)** machinery [144]. Several DNA glycosylases, including hNTH1 (endonuclease III homolog), hOGG1 (8-oxoguanine DNA glycosylase), and NEIL1 (Nei like DNA glycosylase 1), initiate the repair of oxidative DNA damage induced by IR, and this process can lead to their conversion to more dangerous SSBs and DSBs [145]. The essential BER protein apurinic/apyrimidinic endonuclease 1 (APE1) plays a vital role in processing IR-induced oxidative DNA damage and clustered breaks [146]. APE1 deficiency sensitizes cells to IR despite resulting in a smaller number of radiation-induced DSBs, presumably arising when replication forks collide with the SSBs’ intermediates downstream of APE1. Failure to resolve radiation-induced clustered DNA lesions can result in the accumulation of genetically unstable cells [147]. BER can be presumed to play an important role in CIR-induced DNA damage but has not been studied directly in FC. 

Bulky DNA adducts, IR-induced inter-strand crosslinks, cyclopurines, and a variety of oxidative lesions are repaired by the nucleotide excision repair (NER) pathway. Although NER-deficient cells are not hypersensitive to IR, genetic polymorphisms in some NER genes have been associated with radiation-related cancers [148]. Some NER proteins are also involved in the repair of the interstrand crosslinks that IR can induce. IR also induces some types of DNA lesions that can be repaired by NER, such as cyclopurines [149], and a variety of oxidative lesions [150]. Little is known about the role of NER in the response to CIR-induced damage in FC. A study evaluating FCs’ cumulative dose of exposure to CIR (assuming an exposure rate of 6 µSv/hr during flight) showed a significant correlation between the dose of CIR and NER activity in PBMCs measured by the unscheduled DNA synthesis assay [82]. This could reflect the healthy worker effect [21] but would also be consistent with a radio-adaptive response that has been proposed to underlie radiation hormesis [151] and has been observed in other settings [91,152]. 

Additional DNA repair pathways, including Fanconi anemia repair and others, are expected to play a role in the response to CIR-induced DNA damage but require further study. The Fanconi anemia repair pathway eliminates interstrand DNA crosslinks and protects the cells from being killed by IR. It is essential for maintaining the genome’s integrity, and defective repair pathways have been associated with increased cancer risk and immune disorders [153]. In light of the complexity of CIR-induced DNA damage, multiple DNA repair pathways are expected to be involved in its repair. To address this complexity, there is a need to investigate the role of multiple DNA repair mechanisms for CIR-induced DNA damage in FC. Taken together, these results underscore the importance of the involvement of multiple repair pathways in response to CIR, based on the complexity of induced DNA damage and also the need for additional mechanistic studies aimed at understanding the origins of differential levels of sensitivity to high- versus low-LET IR in FC. Such studies are expected to assist in evaluating the potential health effects of in-flight exposure and eventual design strategies to mitigate them.

## 5. Discussion and Conclusions

Due to the complex mixture of mutagenic stressors encountered during air travel, there are many challenges in accurately assessing the exposure of FC to CIR and relating it to the associated health risks. Exposure to CIR changes with altitude and latitude [11,154]. Dosimeters that can provide real-time exposure to CIR levels are not regularly used aboard aircraft during flight. Considering the unique characteristics of CIR and the dose and dose-rate exposure of FC, efforts to link flight-related CIR exposure to health are challenging and perhaps rooted in incorrect assumptions based on our understanding of simple photon radiation. CIR at altitude mainly comprises neutron particles, but the biological effects of neutrons in combination with other exposures to radiation are also understudied. Historically, the overwhelming majority of pilots have been men, but the representation of women has increased drastically in the last decade [155]. Given the risks to reproductive health and development of the fetus in the case of pregnant FC [13], studies focusing on effects of CIR-induced DNA damage in pilots who may become pregnant are particularly needed. Furthermore, although administrative interventions such as schedule rotation can decrease exposure for the individual, the strategy requires a larger pool of workers. As a result, this strategy may not be economically viable.

Radiation-related health outcomes followed in other contexts have often been based on exposure to extremely high acute doses due to single events such as nuclear disasters or detonation of atomic bombs. This risk of radiation exposure in these scenarios differs because of the distinct particle composition and the chronic low-dose and low dose-rate associated with exposures typically experienced during flight. Drawing inferences about the risks associated with flight-related exposure to CIR based on studies of naturally occurring background sources of radiation is subject to similar caveats. From a risk mitigation standpoint, the hierarchy of controls [156] generally used to protect workers from occupational hazards may not apply to FC.

Several epidemiological studies investigating whether cancer risk is associated with exposure to CIR have yielded varying results that might be explained in part by the differences in the studies’ designs and the method of estimating exposure [11]. As a result, the biological mechanism underlying excess cancer risk in FC remains to be determined. Laboratory-based studies focused on the biological effects of the mixed exposures that more accurately reflect the conditions experienced by FC would be valuable. Future studies will be most impactful if careful consideration is given to the choice of the control population to be compared. Because FC must perform physically and emotionally demanding tasks to remain in the workforce, studies comparing FC with reference populations performing other types of work are likely subjected to a form of selection biased known as the “healthy worker effect,” [11,21,157]. The healthy worker effect complicates efforts to estimate the health effects of occupational exposures, because the health status of the workers may be higher than that of the reference group. Thus, if not adequately addressed, the healthy worker effect might be expected to mask the adverse health effects of occupational exposures among FC. 

Since carcinogenesis is driven by genomic instability, we propose a testable model wherein the elevated cancer risk in FC can be explained at least in part by occupational exposure to CIR and other genotoxic agents. Exposure to CIR may promote genomic instability via its direct DNA-damaging effects, and the effects may potentially be enhanced by other exposures of FC that either also damage DNA or lead to the suppression of DNA repair activity (Figure 5). Given the higher abundance of endogenously produced DNA lesions, identifying the CIR-induced DNA damage in FC may require the development of new, highly sensitive, and specific molecular assays. Cytogenetic analysis has provided indirect evidence of CIR-induced genomic instability and is a valuable approach that should be included in future larger studies. PBMCs provide invaluable insights into in vivo human responses to environmental and occupational genotoxic exposures. Emerging high-throughput technologies to quantify DDR such as the CometChip assay [158], rapid automated biodosimetry technology (RABiT) [159], γ-H2AX [159], fluorescence-based multiplexed host cell reactivation (FM-HCR) [29,160,161], and many others can help assess personalized risk and exposures and are compatible with the limited quantity of blood that can usually be obtained from FC study participants [162]. Further improvements to these technologies that enhance the specificity of these assays for CIR-induced biological changes will create new opportunities for their application to answering questions about the biological effects of CIR. These technologies can also potentially identify vulnerable individuals using predictive models or DNA repair biomarkers to assess risk and inform cancer surveillance strategies [29,163]. There is substantial evidence of interindividual variation in DRC correlating to the risk of disease and sensitivity to radiation [57,164,165]. Additional biomarkers of genomic instability such as the length of telomeres [166] and clonal hematopoiesis of indeterminate potential [167] may further clarify the possible role of CIR as an environmental mutagen.

In the future, integrating omics approaches and functional assays together with accurate CIR dosimetry into studies of FC has the potential to reveal whether exposure to CIR causes cancer in FC. These studies may also enable both global and personalized strategies for cancer prevention [162]. Ultimately, the field will require a multidisciplinary approach incorporating epidemiologists, engineers, and mechanistic biologists to effectively measure and study how complex exposure interactions in the real-world flight environment impact the genomic integrity of the FC [15].

## Figures and Tables

**Figure 1 ijms-25-07670-f001:**
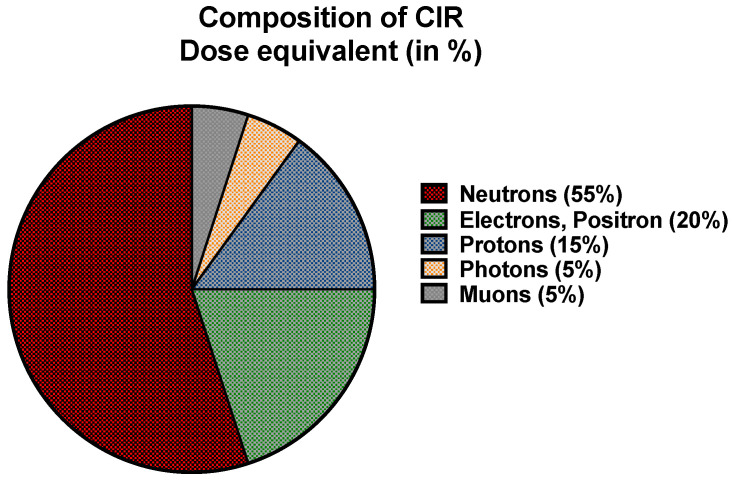
**Components of CIR.** Approximate proportions of the main components of CIR that occur in the atmosphere at aircraft altitudes and temperate latitudes. However, the composition of radiation inside the aircraft is expected to differ [43].

**Figure 2 ijms-25-07670-f002:**
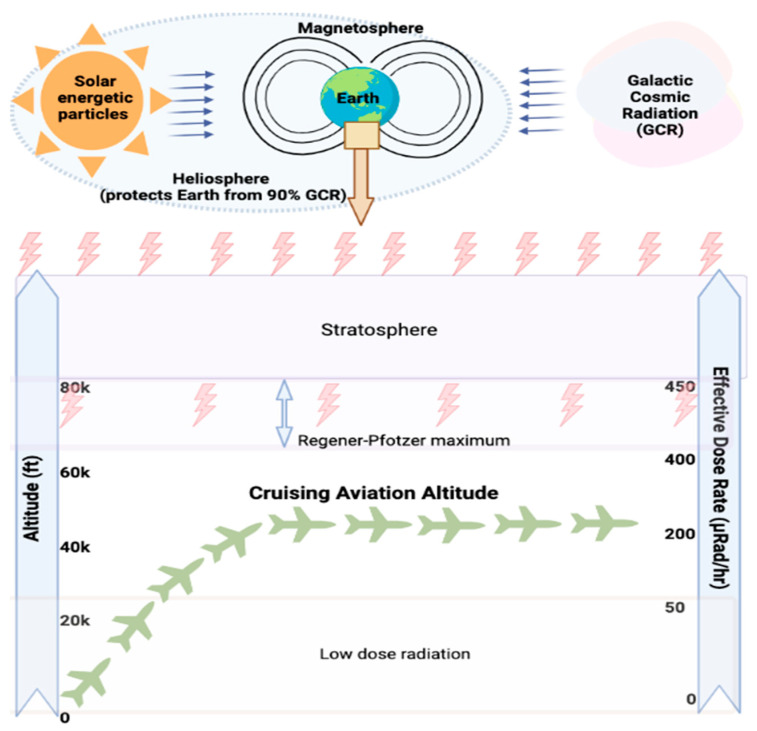
**Exposure to cosmic ionizing radiation (CIR) that occurs at the cruising altitudes of aircraft.** The heliosphere, Earth’s atmosphere, and the magnetic field provide some protection (approx. 90%) from exposure to CIR. On an average, circumpolar flights (shown by the green plane shape) operate at aviation cruising altitudes of 35,000 feet or above. The effective dose rate from exposure to the remaining CIR that passes through the stratosphere (including the peak level of approx. 65,000 feet known as the Regener–Pfotzer maximum) is directly proportional to the cruising altitude of the aircraft. Figure created in BioRender.com accessed on 1 July 2024.

**Figure 3 ijms-25-07670-f003:**
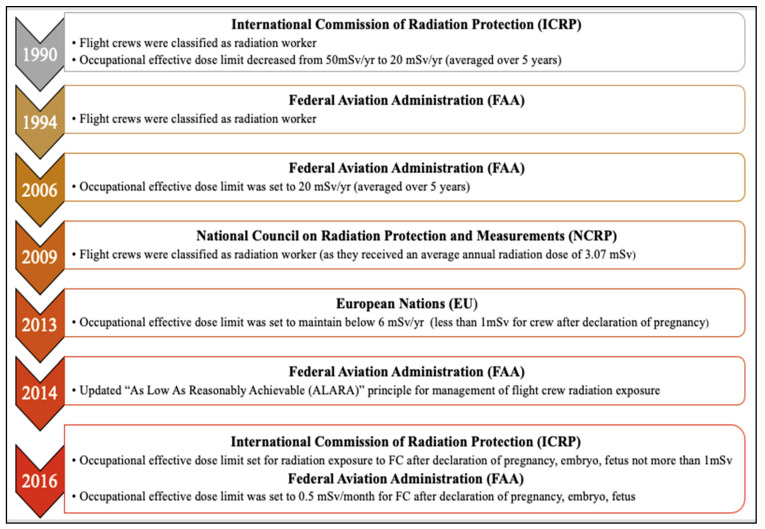
**Regulatory oversight on exposure to cosmic ionizing radiation in flight crews.** Timeline of considerations and amendments made by various regulatory agencies for exposure to occupational radiation in flight crews.

**Figure 4 ijms-25-07670-f004:**
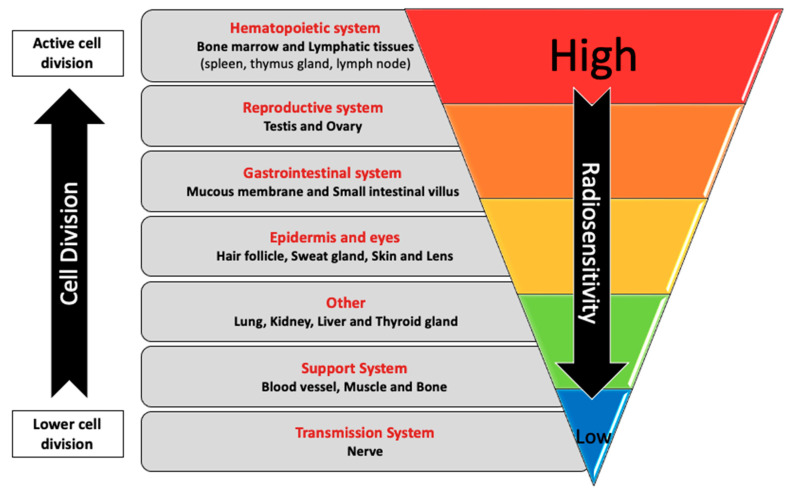
**Radiosensitivity of human tissues and organs.** Cell proliferation is generally inversely related to sensitivity to radiation in human tissues (this figure is adapted from [115], with a modification on the left to reflect the relative frequency of cell division in the various tissues). We note that there are exceptions to these generalizations; for example, while most adult neurons are post-mitotic, some neurogenesis occurs even in adulthood [118,119,120,121].

**Figure 5 ijms-25-07670-f005:**
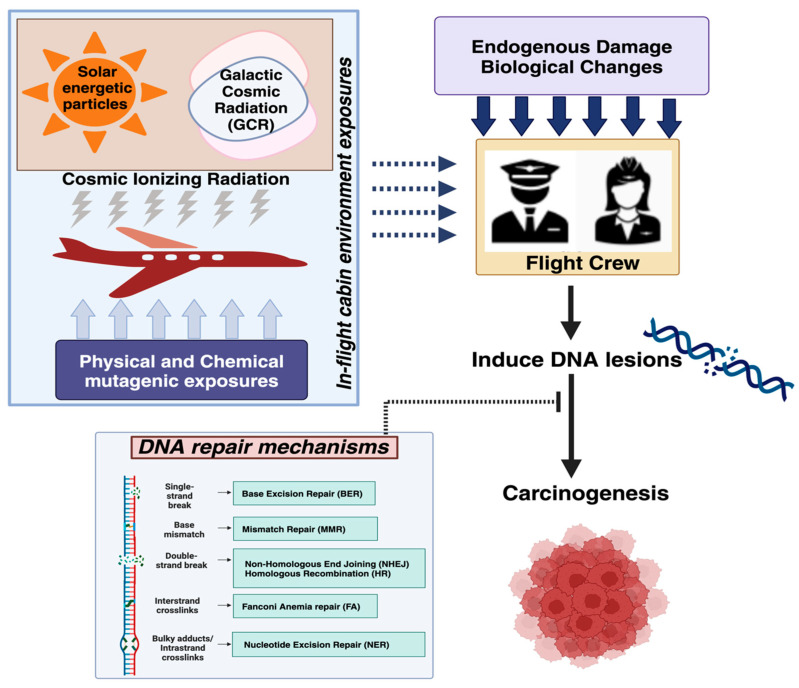
**Proposed model relating air travel, DNA damage and repair, and cancer risk.** FC can be exposed to various DNA-damaging agents during air travel, such damage arising directly (dashed arrow) from flight-related exposures (radiation, physical, and chemicals) and damage arising indirectly (solid arrow) from biological processes that are disrupted during flight (metabolism, respiration, oxidative damage, inflammation, circadian rhythm disruption). Unrepaired DNA damage due to inhibition of or defects in the DNA repair mechanisms (dotted arrow) leads to genomic instability in cells that escape from cell death. Mutations can lead to the initiation of cancer.

**Table 1 ijms-25-07670-t001:** Maximum mean effective limits of ionizing radiation doses by various regulatory bodies * Averaged over 5 years (but not more than 50 mSv in 1 year).

Exposure limits	International Commission of Radiation Protection (ICRP) [23]	European Nations (EU) [43,44]	US National Council on Radiation Protection and Measurements (NCRP) [4]	US Nuclear Regulatory Commission, (NRC) [22]	US Federal Aviation Administration (FAA) [45]
General public	1 mSv/y	1 mSv/y	1 mSv/y	1 mSv/y	1 mSv/y
Pregnant women and unborn fetuses	1 mSv after declaration of pregnancy	1 mSv after declaration of pregnancy	5 mSv and no more than 0.5 mSv in any month	5 mSv and no more than 0.5 mSv in any month	1 mSv and no more than 0.5 mSv in any month
Occupational exposure	20 mSv/y *	20 mSv/y * employer required monitoring and administrative controls to maintain <6 mSv/y	50 mSv/y	50 mSv/y	20 mSv/y * recommendation for FC to self-monitor without requirements for the employer

## Data Availability

The original contributions presented in the study are included in the article; further inquiries can be directed to the corresponding author.

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
