# Peer review of "Cosmic Ionizing Radiation: A DNA Damaging Agent That May Underly Excess Cancer in Flight Crews"

_ijms, 2024, doi:10.3390/ijms25147670_

Round 1

Reviewer 1 Report

Comments and Suggestions for Authors

The manuscript titled “Cosmic ionizing radiation: a DNA damaging agent that may underly excess cancer in flight crew” by Toprani, S.M.; et al. is a Review work where the authors outlined the negative impact of cosmic ionic radiation on the flight crew in terms of cancer incidence. The most relevant bullet points highlighted by the authors could open new gates in the design of the next-generation of cancer therapies exploiting the molecular targets which could benefit not only to pilots and fligth attendants, but also for the rest of society. The manuscript is generally well-written and this is a topic of growing interest.

However, it exists some points that need to be addressed (please, see them below detailed point-by-point) to improve the scientifc quality of the submitted manuscript paper before this article will be consider for its publication in the International Journal of Molecular Sciences.

I)          KEYWORDS. The authors should consider to modify the term “cosmic radiation” by “cosmic ionic radiation” in the keyword list.

II)       INTRODUCTION. “Air travel increases yearly in the United States (U.S), exposing approximately 43,000 pilots and 96,900 flight attendants (FA) (…) at sea level” (lines 27-29). Could the authors provide quantitative data details about the worldwide burdens according to flight crew members? This will significantly aid the potential readers to better understand the significance of this examined field.

III)    CURRENT UNDERSTANDING OF CIR EXPOSURE DURING FLIGHT TRAVEL. In this section, the authors outlined the degree of cosmic ionic radiation exposure depending on the flight altitude. It may be desirable to expand this discussion with the aircraft manufacturing materials and if it exists any customized external coatings to adsorb these detrimental radiation effects and thus, minimizing the exposure degree for the aircraft crew.

IV)    INTER-INDIVIDUAL VARIATION IN RESPONSE TO RADIATION EXPOSURE. This section is clearly explained. No actions are requested from the authors.

V)       DNA DAMAGE AND REPAIR MECHANISMS ASSOCIATED WITH CIR EXPOSURE IN FC. “Each day, cells experience >100,000 spontaneous DNA lesions (…) DNA repair machinery” (lines 290-293). Here, even if I agree with this statement provided by the authors is should be highlighted how the mentioned DNA repair machinery can lead to DNA degradation [1] if it is required for the cell survival and how they can serve as molecular target against cancer diseases [2].

[1] Novo, N.; et al. Beyond a platform protein for the degradosome assembly: The Apoptosis-Inducing Factor as an efficient nuclease involved in chromatinolysis. PNAS Nexus 20222, pgac312. https://doi.org/10.1093/pnasnexus/pgac312.

[2] Siklos, M.; et al. Therapeutic targeting of chromatin: status and opportunities. FEBS J2022289, 1276-1301. https://doi.org/10.1111/febs.15966.

VI)    Then, a schematic representation would be beneficial to highlight the existing DNA reparation mechanism inner the cell (even if some insights are already provided in the Figure 5, line 587).

VII) DISCUSSION AND CONCLUSION. Firstly, the authors should consider to modify the current title by “discussion and conclusions”. Then, this section perfectly remarks the most relevant outcomes found by the authors in this field. The authors should add a brief statement to discuss about the future line actions to pursue this research and the open perspectives.

Comments on the Quality of English Language

The manuscript is generally well-written albeit it may be desirable if the authors could recheck it in order to polish those final details susceptible to be improved.

Author Response

Reviewer 1

1. The manuscript titled “Cosmic ionizing radiation: a DNA damaging agent that may underly excess cancer in flight crew” by Toprani, S.M.; et al. is a review work where the authors outlined the negative impact of cosmic ionic radiation on the flight crew in terms of cancer incidence. The most relevant bullet points highlighted by the authors could open new gates in the design of the next generation of cancer therapies exploiting the molecular targets which could benefit not only to pilots and flight attendants, but also for the rest of society. The manuscript is generally well-written, and this is a topic of growing interest.

Response: We really appreciate the reviewer for the positive constructive feedback on the review.

2. However, it exists some points that need to be addressed (please, see them below detailed point-by-point) to improve the scientific quality of the submitted manuscript paper before this article will be consider for its publication in the International Journal of Molecular SciencesThe authors should consider modifying the term “cosmic radiation” by “cosmic ionic radiation” in the keyword list.

Response: We have changed the keyword from “cosmic radiation” to “cosmic ionizing radiation” in the listed keywords. [Track change/clean copy: Pg 1, Line 32].

3.   INTRODUCTION. “Air travel increases yearly in the United States (U.S), exposing approximately 43,000 pilots and 96,900 flight attendants (FA) (…) at sea level” (lines 27-29). Could the authors provide quantitative data details about the worldwide burdens according to flight crew members? This will significantly aid the potential readers to better understand the significance of this examined field.

Response: According to recent 2023 statistics from IATA, DataUSA.IO, and the Centre for Aviation, there are 351,000 commercial pilots employed globally, with 12.25% (43,000) of them based in the United States. Additionally, there are 400,000 flight attendants worldwide, with 24.2% (96,900) of them working in the U.S. These statistics have been included on page 2, lines 43-46 in track change/clean copy document.

4. CURRENT UNDERSTANDING OF CIR EXPOSURE DURING FLIGHT TRAVEL. In this section, the authors outlined the degree of cosmic ionic radiation exposure depending on the flight altitude. It may be desirable to expand this discussion with the aircraft manufacturing materials and if it exists any customized external coatings to adsorb these detrimental radiation effects and thus, minimizing the exposure degree for the aircraft crew.

Response: Airplanes are not specifically designed with special materials or coatings to protect against cosmic ionizing radiation (CIR). Instead, several other factors contribute to managing radiation exposure during air travel:

  1. Altitude and Flight Path: Radiation exposure is directly related to altitude and latitude. Pilots and flight planners can reduce exposure by adjusting flight paths and altitudes, particularly during solar radiation events.
  2. Aircraft Structure: The aluminum skin and structure of an aircraft provide some level of protection against radiation, though it's not specifically designed for this purpose. This natural shielding is minimal compared to the exposure reduction that could be achieved with specialized materials.
  3. Monitoring and Regulation: Airlines monitor solar activity and cosmic radiation levels. They follow guidelines and regulations set by aviation authorities like the FAA and EASA to ensure radiation exposure remains within safe limits for both crew and passengers.
  4. Operational Procedures: Airlines implement operational procedures such as adjusting flight routes and altitudes during periods of increased solar activity to minimize radiation exposure.
  5. Research and Development: Ongoing research investigates potential materials and methods to enhance radiation protection in aircraft, but as of now, no specific coatings or materials are standard for this purpose.

Overall, while the structural materials of aircraft do offer some degree of natural protection, the main strategies for managing radiation exposure involve operational adjustments and regulatory compliance rather than specialized materials or coatings. But for reader’s understanding, we have covered this in the section 2.1: CIR exposure in flight on Pg 4, line 133-134 in the Track change/clean copy document.

5. INTER-INDIVIDUAL VARIATION IN RESPONSE TO RADIATION EXPOSURE. This section is clearly explained. No actions are requested from the authors. 

Response: We thank the reviewer for the positive feedback.

6. DNA DAMAGE AND REPAIR MECHANISMS ASSOCIATED WITH CIR EXPOSURE IN FC. “Each day, cells experience >100,000 spontaneous DNA lesions (…) DNA repair machinery” (lines 290-293). Here, even if I agree with this statement provided by the authors. It should be highlighted how the mentioned DNA repair machinery can lead to DNA degradation [1] if it is required for the cell survival and how they can serve as molecular target against cancer diseases [2].

[1] Novo, N.; et al. Beyond a platform protein for the degradosome assembly: The Apoptosis-Inducing Factor as an efficient nuclease involved in chromatinolysis. PNAS Nexus 20222, pgac312. https://doi.org/10.1093/pnasnexus/pgac312.

[2] Siklos, M.; et al. Therapeutic targeting of chromatin: status and opportunities. FEBS J2022289, 1276-1301. https://doi.org/10.1111/febs.15966.

Response: We have incorporated the edits as mentioned by the reviewer on Pg 7, line 276 onwards in the Track change/clean copy document.

7. Then, a schematic representation would be beneficial to highlight the existing DNA reparation mechanism inner the cell (even if some insights are already provided in the Figure 5, line 587).

Response: We thank the reviewer for this feedback. We have modified the figure 5 highlighting the naturally existing DNA repair mechanism involved in resolving different type of DNA lesion in the cell. The modified figure can be found on Pg 31 of Track change/clean copy document.

8. DISCUSSION AND CONCLUSION. Firstly, the authors should consider modify the current title by “discussion and conclusions”. Then, this section perfectly remarks the most relevant outcomes found by the authors in this field. The authors should add a brief statement to discuss about the future line actions to pursue this research and the open perspectives.

Response: We acknowledge the reviewer's point regarding the title correction and addressing the future directions for the current research in this area, particularly the lack of biological studies conducted on this specific cohort. To address this, a dedicated section is already existing in the discussion part that outlines these limitations and proposes alternative approaches for future research. For example, when discussing DNA damage and repair responses (the content can be found on Pg 13, line 507 onwards in Track change/clean copy document), where we have suggested specific considerations such as study model selection, dosimeter choice, control population selection, and the potential application of emerging technologies. We have also concluded the discussion the content can be found on Pg 13, line 543 onwards in Track change/clean copy document) by emphasizing the need for a multi-disciplinary approach and collaboration from various experts to effectively address the research hypothesis.

Reviewer 2 Report

Comments and Suggestions for Authors

This is a narrative review elaborating that the Federal Aviation Administration has classified flight crew (FC)—including commercial pilots and flight attendants—as "radiation workers" way back in 1994 due to exposure to cosmic ionizing radiation (CIR). Epidemiological studies show higher cancer incidence and mortality among FC, but the link to CIR remains unconfirmed. The authors reviewed DNA damage and repair related to CIR in FC. However, this review did not provide the latest information. The authors said that “To collect up-to-date information until July 2023, we performed searches on Google and the PubMed website.”(page 2)

The newest data, however, is from a paper (ref 56) published on 20 September 2022.

Also, Figure 4 is misleading. The authors claimed that this figure was adapted from “Individual differences in chromosomal aberrations after in vitro irradiation of cells from healthy individuals, cancer and cancer susceptibility syndrome patients” by Luitpold V.R. Distel et al., which was published on November 20, 2006. https://www.thegreenjournal.com/article/S0167-8140(06)00537-8/fulltext

Figure 4, Radiosensitivity of human tissue and organs. (this figure is adapted from (65)). It portrayed “no cell division” for “nerve”.

As we know, in certain regions of the brain, such as the hippocampus and the olfactory bulb, neurogenesis (the process of generating new neurons) can occur even in adulthood. This is significant for learning, memory, and olfactory function. Besides, unlike neurons, glial cells (which support and protect neurons) retain the ability to divide and proliferate throughout adulthood. These cells include astrocytes, oligodendrocytes, and microglia.

  1. Neurogenesis in the Adult Human Hippocampus:
    • Gage, F. H. (2000). "Mammalian neural stem cells." Science, 287(5457). This review highlights the discovery of neural stem cells and ongoing neurogenesis in the adult hippocampus. https://pubmed.ncbi.nlm.nih.gov/10688783/
    • Eriksson, P. S., et al. (1998). "Neurogenesis in the adult human hippocampus." Nature Medicine, 4(11). This seminal paper provides evidence for the existence of neurogenesis in the adult human hippocampus. https://pubmed.ncbi.nlm.nih.gov/9809557/
  2. Glial Cells and Their Proliferation:
    • Barres, B. A. (2008). "The mystery and magic of glia: A perspective on their roles in health and disease." Neuron, 60(3), 430-440. This review discusses the roles of glial cells and their ability to proliferate in the adult brain. https://pubmed.ncbi.nlm.nih.gov/18995817/
  3. General Overview of Neuronal Cell Division:
    • Kandel, E. R., Schwartz, J. H., & Jessell, T. M. (2013). Principles of Neural Science (5th ed.). McGraw-Hill. This textbook is a comprehensive resource on the functioning of the nervous system, including the capacity for neuronal cell division.

While the majority of neurons do not divide after maturation, the discovery of adult neurogenesis has changed the understanding of the brain's plasticity and its capacity for repair and adaptation.

As this review lacks the latest information, it’s misleading because the most recent studies might correct, refute, or build upon older research. Without the latest references, a review might propagate outdated or incorrect information, leading to inaccuracies in understanding and application.

Author Response

Reviewer 2

This is a narrative review elaborating that the Federal Aviation Administration has classified flight crew (FC)—including commercial pilots and flight attendants—as "radiation workers" way back in 1994 due to exposure to cosmic ionizing radiation (CIR). Epidemiological studies show higher cancer incidence and mortality among FC, but the link to CIR remains unconfirmed. The authors reviewed DNA damage and repair related to CIR in FC. 

1. However, this review did not provide the latest information. The authors said that “To collect up-to-date information until July 2023, we performed searches on Google and the PubMed website”. (page 2). The newest data, however, is from a paper (ref 56) published on 20 September 2022.

Response: We appreciate the reviewer's suggestion to ensure the most up-to-date information is included. To address this, we conducted a comprehensive search for relevant research articles published after July 2023, focusing on DNA damage and repair related to CIR in flight crews. Our search identified a single, pertinent article published in December 2023 (PMID: 38269211, reference 106), which has now been incorporated into the paper (Pg 9, on Line 325-327 and Line 340 in the Track change/clean copy document). Additionally, we have updated the data collection timeframe throughout the manuscript to reflect "until May 2024" for improved accuracy (Pg 3, Line 103 onwards in Track change/clean copy document).

2. Also, Figure 4 is misleading. The authors claimed that this figure was adapted from “Individual differences in chromosomal aberrations after in vitro irradiation of cells from healthy individuals, cancer and cancer susceptibility syndrome patients” by Luitpold V.R. Distel et al., which was published on November 20, 2006. https://www.thegreenjournal.com/article/S0167-8140(06)00537-8/fulltext. Figure 4, Radiosensitivity of human tissue and organs. (this figure is adapted from (65)).It portrayed “no cell division” for “nerve”. As we know, in certain regions of the brain, such as the hippocampus and the olfactory bulb, neurogenesis (the process of generating new neurons) can occur even in adulthood. This is significant for learning, memory, and olfactory function. Besides, unlike neurons, glial cells (which support and protect neurons) retain the ability to divide and proliferate throughout adulthood. These cells include astrocytes, oligodendrocytes, and microglia.

Neurogenesis in the Adult Human Hippocampus:

  • Gage, F. H. (2000). "Mammalian neural stem cells." Science, 287(5457). This review highlights the discovery of neural stem cells and ongoing neurogenesis in the adult hippocampus. https://pubmed.ncbi.nlm.nih.gov/10688783/
  • Eriksson, P. S., et al. (1998). "Neurogenesis in the adult human hippocampus." Nature Medicine, 4(11). This seminal paper provides evidence for the existence of neurogenesis in the adult human hippocampus. https://pubmed.ncbi.nlm.nih.gov/9809557/

Glial Cells and Their Proliferation:

  • Barres, B. A. (2008). "The mystery and magic of glia: A perspective on their roles in health and disease." Neuron, 60(3), 430-440. This review discusses the roles of glial cells and their ability to proliferate in the adult brain. https://pubmed.ncbi.nlm.nih.gov/18995817/

General Overview of Neuronal Cell Division:

  • Kandel, E. R., Schwartz, J. H., & Jessell, T. M. (2013). Principles of Neural Science(5th ed.). McGraw-Hill. This textbook is a comprehensive resource on the functioning of the nervous system, including the capacity for neuronal cell division.

While the majority of neurons do not divide after maturation, the discovery of adult neurogenesis has changed the understanding of the brain's plasticity and its capacity for repair and adaptation.

Response: We thank the reviewer for identifying an inaccuracy in Figure 4. We originally adapted the figure from a source that did not fully reflect current scientific understanding (Ministry of the Environment Government of Japan, 2013) and not Luitpold V.R. Distel et al., 2006. Following the reviewer's insightful comments and supported by research evidence, we have addressed this issue by citing the references in the figure legend. Additionally, we have modified the figure to depict lower cell proliferation activity at the lower end, rather than no cell division, as neurogenesis is indeed a process that occurs in the adult brain. The following edits can be found on Figure 4, Pg 30 in Track change/ clean copy document.

3. As this review lacks the latest information, it’s misleading because the most recent studies might correct, refute, or build upon older research. Without the latest references, a review might propagate outdated or incorrect information, leading to inaccuracies in understanding and application.

Response: Thank you for your valuable feedback regarding the importance of incorporating the latest information. We completely agree that including recent studies strengthens the review and minimizes the risk of presenting outdated information. To address this, we have reviewed and incorporated relevant findings from Google and PubMed website until May 2024 as stated in our response 1. We believe this strengthens the review keeping the discussion unchanged by highlighting the potential new directions needed to address the effects of CIR on FC. We have carefully ensured that the core understanding and application of the reviewed topic remain consistent.
